# Discovery of New Anti-Cancer Agents against Patient-Derived Sorafenib-Resistant Papillary Thyroid Cancer

**DOI:** 10.3390/ijms242216413

**Published:** 2023-11-16

**Authors:** Yuna Kim, Hyeok Jun Yun, Kyung Hwa Choi, Chan Wung Kim, Jae Ha Lee, Raymond Weicker, Seok-Mo Kim, Ki Cheong Park

**Affiliations:** 1Department of Internal Medicine, Institute of Gastroenterology, Gangnam Severance Hospital, Yonsei University College of Medicine, 211 Eonjuro, Gangnam-gu, Seoul 06273, Republic of Korea; sadts@yuhs.ac; 2Department of Surgery, Thyroid Cancer Center, Institute of Refractory Thyroid Cancer, Gangnam Severance Hospital, Yonsei University College of Medicine, Seoul 06273, Republic of Korea; gsyhj@yuhs.ac; 3Department of Urology, CHA Bundang Medical Center, CHA University, Seongnam 13496, Republic of Korea; khchoi@cha.ac.kr; 4CKP Therapeutics, Inc., 110 Canal Street, Lowell, MA 01852, USA; ckim@ckptherapeutics.com (C.W.K.); jlee@ckptherapeutics.com (J.H.L.);; 5Department of Surgery, Yonsei University College of Medicine, 50-1, Yonsei-ro, Seodaemun-gu, Seoul 03722, Republic of Korea

**Keywords:** patient-derived papillary thyroid cancer, sorafenib, anti-cancer drug resistant papillary thyroid cancer

## Abstract

Thyroid cancer is the most well-known type of endocrine cancer that is easily treatable and can be completely cured in most cases. Nonetheless, anti-cancer drug-resistant metastasis or recurrence may occur and lead to the failure of cancer therapy, which eventually leads to the death of a patient with cancer. This study aimed to detect novel thyroid cancer target candidates based on validating and identifying one of many anti-cancer drug-resistant targets in patient-derived sorafenib-resistant papillary thyroid cancer (PTC). We focused on targeting the sarco/endoplasmic reticulum calcium ATPase (SERCA) in patient-derived sorafenib-resistant PTC cells compared with patient-derived sorafenib-sensitive PTC cells. We discovered novel SERCA inhibitors (candidates 33 and 36) by virtual screening. These candidates are novel SERCA inhibitors that lead to remarkable tumor shrinkage in a xenograft tumor model of sorafenib-resistant patient-derived PTC cells. These results are clinically valuable for the progression of novel combinatorial strategies that facultatively and efficiently target extremely malignant cancer cells, such as anti-cancer drug-resistant PTC cells.

## 1. Introduction

Thyroid cancer (TC) is a well-known disease wherein malignant cancer cells form in the thyroid gland on the base of the neck. It is customarily categorized into four cancer subtypes: papillary (PTC), follicular (FTC), medullary (MTC), and anaplastic (ATC) [1,2,3,4]. The categorization of TC provides a basis for the histological type of thyroid cells. De-differentiated thyroid cancer (DeTC) and differentiated thyroid cancer (DTC) arise from follicular cells. Differentiated thyroid cancer is sub-classified into PTC, FTC, and Hürtle cell carcinoma (HCC). In particular, TC is categorized into well-differentiated and un-differentiated classes in relation to clinical disclosure based on conventional classification [5,6]. Well-differentiated thyroid cancer (WDTC) usually indicates a good prognosis that is curable, whereas poorly differentiated thyroid cancer (PDTC) and involved undifferentiated thyroid cancer (UTC) have an infrequent and aggressive prognosis [7,8]. Anti-cancer drug-resistant TC is a PDTC that leads to patient death through recurrence or metastasis [9,10,11]. Clinical behavior based on the molecular and biological mechanisms of anti-cancer drug-sensitive cancer and drug-resistant cancer is distinct [12,13]. Many researchers have focused on making a difference by analyzing the span of mutations between anti-cancer drug-sensitive cancer and drug-resistant cancer [14,15,16,17,18,19,20]. However, the mechanism of the molecular distinctions to fully determine the anti-cancer drug-resistant-mediated poor prognosis in patients with PTC is undetermined. Characteristics of these refractory PTC continuously gained anti-cancer drug resistance; therefore, there is a requirement for effective and novel clinical approaches [21,22,23]. Sarco/endoplasmic reticulum calcium ATPase (SERCA) is a key regulator of cytosolic free calcium. Current research showed many dominantly expressed targets of SERCA in patient-derived sorafenib-resistant PTC but not in patient-derived sorafenib-sensitive PTC [24,25,26]. Administration of cytoplasmic free calcium is involved in many cellular procedures involving cellular death of survival known as apoptosis and autophagy under severe endoplasmic reticulum (ER) stress conditions [26,27].

Here, we identified and validated new SERCA inhibitors that are potential therapeutic agents for the anti-cancer drug sorafenib-resistant PTC. The results of this research would help advance innovative target-mediated combinatorial schemes by selectively and effectively targeting refractory cancer cells, such as sorafenib-resistant cancer cells.

## 2. Results

### 2.1. Characteristics and Information of Patient-Derived PTC Cell Lines

Current research has recruited three types of PTC cell lines obtained from patient specimens (Table 1): YUMC-S-P1 (first acquired patient-derived sorafenib-sensitive PTC), YUMC-R-P5 and -P6 (fifth and sixth isolated from patient specimen with sorafenib-resistant mediated recurrence or metastasis) are classified as patients with PTC therapy at the Severance Hospital, Yonsei University College of Medicine, Seoul, Republic of Korea. YUMC-R-P5 and -P6 sorafenib-resistant PTC was more refractory than sorafenib-sensitive PTC, YUMC-S-P1, and metastasis or recurrence were indicated in these patients (Table 1).

### 2.2. A Dissimilarity of Genetic Change and Stimulated Signaling Pathways between Patient-Derived Sorafenib-Sensitive and Sorafenib-Resistant PTC Cell Lines

We performed RNA sequencing (RNA-Seq) based on transcriptome analysis to estimate the stimulated signaling pathway and genetic alterations between sorafenib-sensitive PTC (YUMC-S-P1) and sorafenib-resistant PTC (YUMC-R-P5 and -P6) (Figure 1A–C). YUMC-R-P5 and -P6 cells were implicated with a drastic increase in the EMT markers *ZEB* (*Zinc finger E-box-binding homeobox*), *TWIST* (*twist family bHLH transcription factor*), and *SNAIL* (*Zinc finger protein SNAI1*) compared with YUMC-S-P1, sorafenib-sensitive PTC (Figure 1A). Sorafenib-resistant PTC showed the most noticeable variation of the cancer stemness marker (*KRT17^high^*, *KRT19^low^*, *ALDH1A1^high^*, *PROM1[CD133]^high^*, *CD44^high^*, *CD24^low^*, and *SOX2^high^*), FGF/FGFR (*Fibroblast growth factor/Fibroblast growth factor receptors*, *FGF1*, *FGF5*, *FGF11*, *FGF13*, *FGF16*, *FGFR2*, *FGFR3*, and *FGFR4*), and EMT (*SNAIL1*, *SNAIL2*, *ZEB1*, *ZEB2*, and *TWIST1*) (Figure 1B, top, middle, and bottom). We have particularly focused on the results showing that sorafenib-resistant PTC was greatly stimulated by calcium, and cancer stemness involved the following signaling pathways (calcium, Notch, Wnt, PPAR, PI3K/Akt, and TGF/SMAD) than sorafenib-sensitive PTC (Figure 1C, top and bottom) in Kyoto Encyclopedia of Genes and Genomes (KEGG) pathway analysis [28,29,30,31,32]. We hypothesized that highly stimulated calcium-related signaling pathways in sorafenib-resistant PTC cells were key players in escaping cytoplasmic calcium mediated apoptosis under severe endoplasmic reticulum (ER) stress conditions via anti-cancer drug-treated conditions [28,29,33,34]. RNA-seq analysis showed that there was a minor distinction between *SERCA* (*ATP2A*) isoforms. Basal levels of *SERCA* expression (Primer sequences were indicated Appendix A) were raised in YUMC-R-P5 and -P6 sorafenib-resistant PTC cells compared with sorafenib-sensitive PTC, YUMC-S-P1 (Figure 1D). SERCA is a key regulator of calcium homeostasis to escape anti-apoptosis under severe ER stress conditions [25,35,36,37].

Altogether, sorafenib-resistant PTC might be of prominent value as a therapeutic approach to regulate the metastasis or recurrence in patients with a refractory PTC subtype. These results proposed that regulation of SERCA expression in sorafenib-resistant PTC cells was one of the crucial factors in prolonging survival under sorafenib-treated conditions.

### 2.3. SERCAs Being Key Regulators to Prolong Survival under Sorafenib-Treated Conditions in Sorafenib-Resistant PTC Cells

SERCA is a key regulator of cytoplasmic calcium-mediated apoptosis under severe ER stress conditions [25,35,36,37]. YUMC-R-P5 and -P6 cells in sorafenib-resistant PTC had highly elevated SERCA1 levels among the other SERCA isoforms treated with sorafenib compared to sorafenib-sensitive PTC YUMC-S-P1 cells (Figure 2A). We performed a cell viability assay in the presence of a calcium channel blocker (Nifedipine), an NCX inhibitor (KB-R7943), a PMCA inhibitor (Caloxin2a1), and a SERCA inhibitor (thapsigargin) alone or combined with sorafenib to demonstrate that the key regulator for extended survival under severe ER stress conditions via treatment with sorafenib in sorafenib-resistant PTC cells was SERCA and not NCX (Na^+^/Ca^2+^ exchanger) or calcium ion channels (Figure 2B,C). Calcium channel blockers, NCX, PMCA, and SERCA inhibitors treated without sorafenib in YUMC-R-P5 and -P6 cells showed no considerable changes, while combined treatment with only a SERCA inhibitor (Thapsigargin) and sorafenib failed to survive in a dose-dependent manner (Figure 2B,C). In addition, SERCA expression was highly increased in the presence of sorafenib compared with that without sorafenib in sorafenib-resistant PTC. The induction of SERCA expression and CHOP (an ER stress marker of thapsigargin) in the SERCA inhibitor-treated group was significantly increased compared with the calcium channel blocker (Nifedipine), NCX inhibitor (KB-R7943), and PMCA inhibitor (Caloxin 2a1) (Figure 2D). Sorafenib contributed to SERCA expression, unlike its absence. Inhibition of SERCA by SERCA inhibitors increased CHOP levels, regardless of increased SERCA expression. Combined sorafenib and calcium channel blockers, NCX inhibitors, or PMCA inhibitors showed no significant increase in ER stress. These results proved that SERCA was a critical player in regulating overloaded cytoplasmic free calcium under severe ER stress conditions through treatment with sorafenib.

Taken together, SERCA was a logical target for escaping cytoplasmic calcium mediated apoptosis under severe ER stress conditions via sorafenib treatment in sorafenib-resistant PTC cells.

### 2.4. SERCA Target Specific Discovery of the New Therapeutic Access of Candidates 33 and 36 by In-Silico Screening for Suppression of Patient-Derived Sorafenib-Resistant PTC

We hypothesized that the functional restraint of SERCA could be a practicable therapeutic approach for sorafenib-resistant PTC cells on the basis of results from patient-derived sorafenib-resistant PTC cells. We investigate chemical compounds capable of binding to SERCA and possible pharmacophoric binding modes through in silico screening analysis. Consequently, 1180 (based on docking score), 95 (manually chosen), and 15 candidate (natural product encompassed) compounds were distinguished. Among these, two candidates (33 and 36) showed relatively high binding affinity to SERCA and significantly inhibited SERCA (Figure 3A). Candidates 33 and 36 present an original therapeutic approach for sorafenib-resistant PTC with sorafenib as an anti-cancer drug. Moreover, we performed cell viability assays and immunoblot analysis between sorafenib-sensitive (YUMC-S-P1) and -resistant PTC (YUMC-R-P5 and -P6) cells in the presence of sorafenib alone or with SERCA inhibitors, including novel candidates 33 or 36, to identify the anti-cancer effect of candidates 33 and 36 (Figure 3B,C). The viability of sorafenib-sensitive PTC cells (YUMC-S-P1) was remarkably decreased in a dose-dependent manner following sorafenib treatment with or without candidates 33 or 36 (Figure 3B, left). The viability of sorafenib-sensitive PTC decreased with sorafenib, regardless of the presence or absence of SERCA inhibitors. Meanwhile, the viability of sorafenib-resistant PTC cells (YUMC-R-P5 and -P6) was not considerably influenced by sorafenib. However, SERCA inhibitors (positive control) or candidates 33 or 36 (novel candidates of SERCA inhibitors) combined with sorafenib groups noticeably suppressed the viability of sorafenib-resistant PTC cells in a dose-dependent manner (Figure 3B, middle and right). SERCA inhibitors (Thapsigargin, candidates 33 and 36) alone did not considerably influence the viability of sorafenib-sensitive and drug-resistant PTC cells. The half-maximal inhibitory concentration (IC_50_) of the sorafenib treatment alone was 22 µM in sorafenib-sensitive PTC cells (Table 2). There is no notable difference in the IC_50_ of sorafenib alone or combined with SERCA inhibitors during treatment. Meanwhile, the anti-cancer influence of sorafenib treatment alone showed no meaningful point in sorafenib-resistant PTC. However, the anti-cancer influence of sorafenib was extremely strong when combined with SERCA inhibitors. The IC_50_ of sorafenib was approximately 22–23 µM in YUMC-R-P5 and -P6 when combined with SERCA inhibitors (thapsigargin or candidates 33 and 36) (Table 2).

Taken together, the current results showed that SERCA could be an essential factor prolonging survival through regulation of overloaded cytoplasmic free calcium under severe ER stress conditions by treatment with sorafenib in sorafenib-resistant PTC cells.

### 2.5. New Clinical Approach for Targeted Therapy through Novel Candidates 33 and 36 in a Patient-Derived Sorafenib-Resistant PTC Cell Mouse Xenograft Model

We performed a mouse xenograft model with sorafenib-sensitive (YUMC-S-P1) and -resistant PTC (YUMC-R-P5 and -P6) cell treatment with sorafenib alone or combined with SERCA inhibitors (thapsigargin, candidates 33 and 36) to estimate the combinatorial anti-cancer influence of novel SERCA inhibitors candidates 33 and 36 with sorafenib. We performed oral administration of sorafenib to measure cytotoxic stress in the mouse xenograft model. Tumor shrinkage significantly increased in the sorafenib treatment group regardless of the administration of SERCA inhibitors (Figure 4A, top) in a sorafenib-sensitive PTC cell xenograft model. Significant changes were observed in the sorafenib-resistant PTC xenograft model following sorafenib treatment. Meanwhile, combined treatment with sorafenib and SERCA inhibitors (thapsigargin, novel candidates 33 or 36) resulted in significant tumor shrinkage (Figure 4B, top). The resected tumor weight was not much different from the consequence of the change in tumor volume. The resected tumor weight of sorafenib-sensitive PTC significantly declined in the presence of sorafenib (Figure 4A, middle). There was no meaningful distinction between sorafenib treatment with and without SERCA inhibitors. Sorafenib alone did not significantly reduce tumor weight in sorafenib-resistant PTC, while combined treatment with sorafenib and SERCA inhibitors (thapsigargin, novel candidate 33 and 36) groups markedly decreased it (Figure 4B,C, middle). Singular treatment with each agent did not noticeably affect the overall body weight (Figure 4A–C, bottom). We performed immunoblot assays of tumor tissues to confirm the relationship between SERCA expression and ER stress under severe ER stress conditions with the anti-cancer drug sorafenib. SERCA expression was not meaningfully altered following treatment with each agent alone or combined with sorafenib and SERCA inhibitors in sorafenib-sensitive PTC (Figure 5A). Sorafenib significantly increased the expression of CHOP (the ER stress marker) when administered alone or combined with SERCA inhibitors (Figure 5A). However, sorafenib-resistant PTC highly expressed SERCA and was treated with sorafenib alone or combined with SERCA inhibitors (Figure 5B,C). Nevertheless, highly expressed SERCA in sorafenib-resistant PTC was caused by an increase in ER stress by novel SERCA inhibitors (candidates 33 and 36) via functional inhibition of SERCA (Figure 5B,C).

Overall, these results suggested that novel SERCA inhibitors (candidates 33 and 36) could be a new therapeutic approach for noticeable tumor shrinkage in a xenograft tumor model of patient-derived anti-cancer drug resistant PTC.

## 3. Discussion

Papillary thyroid cancer is a commonplace endocrinological carcinoma with infrequent anti-cancer drug-resistant metastasis or recurrence, showing a poor prognosis that may result in death. However, instances of anti-cancer drug-resistant-mediated metastasis or recurrent PTC, though infrequent, are associated with poor prognosis, potentially leading to patient mortality [38,39]. Moreover, anti-cancer drug-resistant-mediated metastasis or recurrent PTC has demonstrated intractability to most medical remedies [8,10,40,41,42]. Refractory and aggressive PTC initially grows slowly, but it can progress to poorly or undifferentiated cancer, characterized by rapid enlargement and a poor prognosis [43,44]. Variable oncogenic mechanisms and cytogenetic incidents occur in the oncogenesis of refractory TC [45,46,47,48]. A natural consequence of aggressiveness for cancer stemness in refractory PTC is not clearly revealed. Outcomes from the advancement of anti-cancer drug discovery demonstrated that preoperative chemotherapy increased the survival ratio after operation, and most studies have proved the effectiveness of rationally performed chemotherapy and surgery even when efficacious treatment was regarded as unrealizable. Numerous studies have confirmed the effectiveness of well-planned chemotherapy and surgical intervention, even in cases where efficacious treatment was previously deemed unattainable [49,50,51,52]. Unfortunately, no alternative medical options were received as the neo-adjuvant or basal-adjuvant background for drug-resistant cancer [53,54,55], and mass patients with drug-resistant cancer died. Consequently, unmet medical needs were steadily demanded. Drug-resistant cancer is a conclusive part of unmet medical needs [56,57,58,59,60] that remain attached to an essential sticking point for therapeutic solutions for patients with refractory cancer [61,62]. Therefore, the discovery of hit-to-lead was meant to challenge cancer therapy based on the mechanism of anti-cancer drug resistant cancer for suppressing refractory cancer [19,62,63,64].

This study discovered and proposed novel candidates based on mRNA-Seq analysis between patient-derived sorafenib-sensitive and resistant PTC cells to help treat refractory cancer. In particular, we concentrated on high-ranking calcium and Notch signaling pathways among 15 meaningfully increased signaling pathways in patient-derived sorafenib-resistant PTC cells compared with patient-derived sorafenib-sensitive PTC cells. The interrelation between calcium and the Notch signaling pathway is well studied [65,66,67]. More importantly, notch signaling is regulated by SERCA regulation [68,69]. Taken together, we primarily concentrated on crucial genes and signaling pathways of calcium homeostasis to prolong survival under severe ER stress conditions through anti-cancer drug treatment in sorafenib-resistant PTC cells. The final goal of the current research is to propose that novel SERCA inhibitors (candidates 33 or 36) could cause severe ER stress-mediated apoptosis. These candidates meaningfully induced death by functional inhibition of SERCA under severe ER stress conditions. This finding could be favorable to establishing prospective, reasonable therapeutic approaches in patients with refractory PTC to advance efficacious therapy. Further investigation is required to establish the current therapeutic approach. Furthermore, more study is required owing to the limitations of several patient results. In spite of these limitations, this study suggests that the therapeutic solution using sorafenib and SERCA inhibitors (candidates 33 and 36) can treat patient-derived sorafenib-resistant PTC.

## 4. Materials and Methods

### 4.1. Study Design and Ethical Considerations

This study was a retrospective, solitary center analysis of patients with PTC; detailed information was described in our previous study [30,70,71] and Supplementary Methods. This study protocol was confirmed by the Institutional Review Board (IRB) of Severance Hospital, Yonsei University College of Medicine (IRB protocol: 3-2022-0331). Cell samples were isolated from patient specimens at the Severance Hospital, Yonsei University College of Medicine, Seoul, Republic of Korea.

### 4.2. Patients

#### 4.2.1. Patient 1

YUMC-S-P1 was 53 years old and a man with papillary thyroid cancer. Detailed information was described in Supplementary Methods. Currently, radiologic examination and thyroid hormone tests are being followed without recurrence.

#### 4.2.2. Patient 2 and 3

YUMC-R-P5 and -P6 were 52 and 57-year-old women and men with papillary thyroid cancer. Detailed information was described in Supplementary Methods. This patient was treated with sorafenib, after which the disease progression was confirmed in the sorafenib response assessment. Currently, cancer recurrence and metastasis were caused and confirmed after sorafenib was prescribed.

### 4.3. Patient Tissue Specimens

A fresh tumor specimen was dissected from patients with biochemical and histologically proven PTC who were cured at the Severance Hospital, Yonsei University College of Medicine, Seoul, Republic of Korea. Fresh tumors were collected throughout the surgical excision of PTC metastatic and primary sites.

### 4.4. Primary Culture and Cancer Cell Isolation

After resection, tumors were kept in phosphate-buffered saline (PBS) with antifungals and antibiotics and moved to the laboratory. Normal tissue and fat were eliminated and rinsed with 1× Hank’s Balanced Salt Solution. YUMC-S-P1, YUMC-R-P5, and -6 were obtained from papillary thyroid cancer patients treated at the Severance Hospital, Yonsei University College of Medicine, Seoul, Republic of Korea. Tumors were minced in a tube with dissociation medium containing DMEM/F12 and 20% fetal bovine serum supplemented with 1 mg/mL collagenase type IV (Sigma, St. Louis, MO, USA; C5138). The isolated cancer cells were grown in Roswell Park Memorial Institute-1640 medium, supplemented with 15% fetal bovine serum and antibiotics (penicillin and streptomycin). Mycoplasmal contamination was checked for with the Lookout Mycoplasma PCR Detection Kit (Sigma-Aldrich, St. Louis, MO, USA; MP0035).

### 4.5. mRNA-Seq Data

We preprocessed the raw reads from the sequencer to remove low-quality and adapter sequences before analysis and aligned the processed reads to Homo sapiens (GRCh37) using HISAT v2.1.0. The detailed protocol can be found in Supplementary Methods.

### 4.6. Statistical Analysis of Gene Expression Level

The relative abundances of genes were measured in Read Count using StringTie. We performed statistical analyses to find differentially expressed genes using the estimates of abundances for each gene in the samples. Genes with one more than zero Read Count value in the samples were excluded. To facilitate log2 transformation, 1 was added to each Read Count value of filtered genes. Filtered data were log2-transformed and subjected to trimmed mean of M-values (TMM) normalization. The statistical significance of the differential expression data were determined using exactTest, edgeR, and fold change, in which the null hypothesis was that no difference exists among groups. The False discovery rate (FDR) was controlled by adjusting the *p*-value using the Benjamini-Hochberg algorithm. For DEG sets, hierarchical clustering analysis was performed using complete linkage and Euclidean distance as a measure of similarity. Gene enrichment, functional annotation analysis, and pathway analysis for significant gene lists were performed based on Gene Ontology and KEGG pathway analyses.

### 4.7. Hierarchical Clustering

Hierarchical clustering analysis was carried out with complete linkage and Euclidean distance as a measure of resemblance to indicate the expression patterns of dissimilarly indicated transcripts that are satisfied with | fold change | ≥ 2 and an independent *t*-test with a raw *p* < 0.05. All data analysis and visualization of dissimilarly indicated genes were directed at R 3.5.1 (www.r-project.org, accessed on 16 August 2023).

### 4.8. Cell Viability Assay

Cell viability was calculated by the MTT (3-(4,5-Dimethylthiazol-2-yl)-2,5-Diphenyltetrazolium Bromide) assay. Cells were seeded in 96-well plates at 8 × 10^3^ cells per well and cultured overnight to achieve over 80% confluency. Cells were incubated for the indicated times prior to the determination of cell proliferation using the MTT reagent (Roche, Basel, Switzerland; 11465007001) according to the manufacturer’s protocol. Absorbance was measured at 550 nm. Viable cells were counted by trypan blue exclusion. Data were indicated as a percentage of the signal observed in vehicle-treated cells and are shown as the means ± SEM of triplicate experiments.

### 4.9. Immunoblot Analysis

The primary antibodies sarco/endoplasmic reticulum calcium ATPase SERCA1 (1:500, abcam, Cambridge, UK, #133275), SERCA2 (1:500, abcam, #137020), SERCA3 (1:300, abcam, #154259), C/−EBP homologous protein (CHOP, 1:100, Santa Cruz Biotechnology, Santa Cruz, CA, USA, #7351), Bcl-2 (1:500, Cell Signaling Technology, Danvers, MA, USA, #4223S), caspase-3 (1:500, Cell Signaling Technology, Beverly, MA, USA, #9661), and β-actin (1:2000, Santa Cruz Biotechnology, Santa Cruz, CA, USA, #47778) were purchased and maintained overnight at 4 °C. Equal amounts of protein were separated on 8–10% sodium dodecyl sulfate-polyacrylamide gels; the resolved proteins were electro-transferred onto polyvinylidene fluoride membranes (Millipore, Bedford, MA, USA). The membranes were subsequently blocked with 5–10% nonfat milk in TBST for 1 h at room temperature or overnight at 4 °C and incubated with appropriate concentrations of primary antibodies overnight at 4 °C. The membranes were then rinsed 3–10 times with TBST and probed with the corresponding secondary antibodies conjugated to horse radish peroxidase (Santa Cruz) at room temperature for 1 h. After rinsing, the blots were developed with ECL reagents (Pierce) and exposed using Kodak X-OMAT AR Film (Eastman Kodak, Rochester, NY, USA) for 1–5 min. Nuclear fractions were prepared using the NE-PER Nuclear and Cytoplasmic Extraction reagents (Thermo Fisher Scientific, Waltham, MA, USA; #78833) in accordance with the manufacturer’s instructions. Separated nuclear and cytoplasmic extracts were isolated with a protein extraction solution (PRO-PREP, iNtRON Biotechnology, Seoul, Republic of Korea, #17081) or histone extraction kit (Abcam, Cambridge, UK, #113476). Protein extracts and bands were quantified using NanoDrop 2000 (Thermo Fisher Scientific, Waltham, MA, USA) and ImageJ software (NIH, Bethesda, MD, USA; https://imagej.nih.gov, 10 May 2023). The detailed protocol can be found in our previous article [30,32,70,71].

### 4.10. Used Information of Calcium Channel, NCX, PMCA and SERCA Inhibitors

The used doses of nifedipine (Sigma-Aldrich, St. Louis, MO, USA; #21829-25-4), KB R7943 (Sigma-Aldrich, St. Louis, MO, USA; #182004-64-4), caloxin2a1 (AnaSpec, Fremont, CA, USA, #AS-62604), and thapsigargin (Thermo Fisher Scientific, Waltham, MA, USA, #67526-95-8) were 25 μM. All agents were soluble in DMSO.

### 4.11. Human PTC Cell Xenograft

All experiments were approved by the Animal Experiment Committee of Yonsei University. YUMC-S-P1, YUMC-R-P5, and -P6 patient-derived PTC cells (5.2 × 10^6^ cells/mouse) were cultured in vitro and then injected subcutaneously into the upper left flank region of female NOD/Shi-scid, IL-2Rγ KOJic (NOG) mice. After 14 days, tumor-bearing mice were grouped randomly (*n* = 10 per group) and treated with 25 mg/kg SERCA inhibitors, thapsigargin, candidates 33 and 36 (p.o.; per os, oral administration), and 80 mg/kg sorafenib (p.o.) either alone or in combination (excluded for combination of SERCA inhibitors). All agents were diluted in normal saline, and maximam volueme was administered in one dose, up to less than 200 μL. The tumor size was measured every three days using calipers. Tumor volume was estimated using the following formula: L × S2/2 (L, longest diameter; S, shortest diameter). Animals were maintained under specific pathogen-free conditions, and all experiments were approved by the Animal Experiment Committee of Yonsei University (IACUC approval No. 2022-0105).

### 4.12. Statistical Analysis

Statistical analyses were performed using GraphPad Prism 6.0 software (GraphPad Software, La Jolla, CA, USA), Microsoft Excel (Microsoft Corp., Redmond, WA, USA), and R version 2.17. A one-way ANOVA was performed for the multi-group analysis, and a two-tailed Student’s *t*-test was performed for the two-group analysis. Values were expressed as mean ± standard error of mean. *p* values < 0.05 were considered statistically significant.

## 5. Conclusions

SERCA was responsible for cellular-resistant cytotoxic stress under sorafenib-treated conditions. We showed that SERCA inhibitors (candidates 33 and 36) combined with sorafenib induce tumor shrinkage in in vitro and in vivo models of patient-derived sorafenib-resistant PTC cells. This novel combinatorial scheme targeted the effective therapy of incredibly malignant anti-cancer drug-resistant cancer cells.

## Figures and Tables

**Figure 1 ijms-24-16413-f001:**
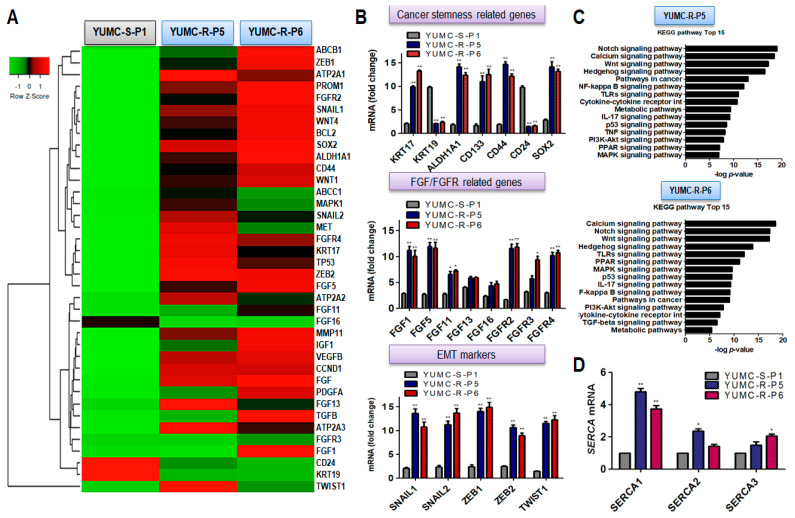
Properties of all currently studied patient-derived papillary thyroid cancer (PTC) cell lines. (**A**) Dissimilarity in gene expressions was indicated through hierarchical clustering; there was a transition in the gene expression profile between patient-derived sorafenib-sensitive and resistant PTC cells. (**B**) Analysis for gene expression variance set up on mRNA seq analysis for cancer stem cell (CSC) markers, fibroblast growth factor (FGF)/FGF receptor-related genes, and epithelial-mesenchymal transition (EMT) markers between patient-derived sorafenib-sensitive and -resistant PTC cells. All data were normalized to α-tubulin expression. (**C**) Bar plot exposing 15 markedly stimulated pathways in sorafenib-resistant PTC cells, YUMC-R-P5 (top) and YUMC-R-P6 (bottom). (**D**) Transition of SERCA isoform-dependent RNA expression between sorafenib-sensitive and -resistant PTC cells under basal conditions. * *p* < 0.05 vs. sorafenib-sensitive PTC cells, YUMC-S-P1, ** *p* < 0.01 vs. sorafenib-sensitive PTC cells, YUMC-S-P1.

**Figure 2 ijms-24-16413-f002:**
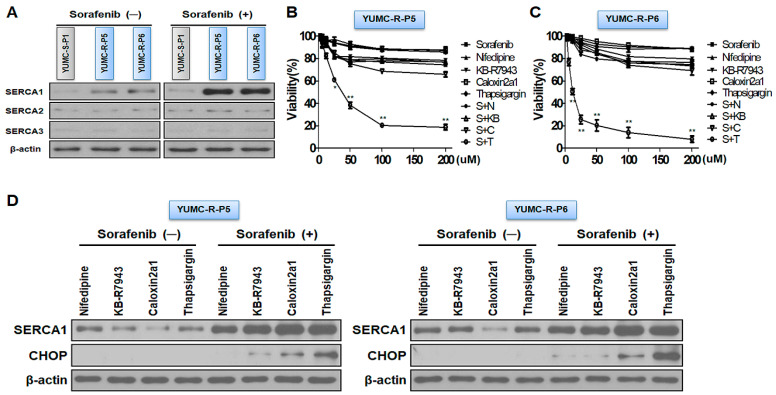
SERCA is a key regulator for prolonged survival under severe endoplasmic reticulum (ER) stress conditions through sorafenib-treated conditions. (**A**) Immunoblot analysis measuring changes in SERCA isoforms with or without sorafenib between patient-derived sorafenib-sensitive and -resistant PTC cells. (**B**,**C**) Cell viability assay showing dose-dependent additions of calcium channel blocker (Nifedipine), Na^+^/Ca^2+^ exchanger (NCX) inhibitor (KB-R7943), PMCA inhibitor (Caloxin 2a1), and SERCA inhibitor (Thapsigargin) alone or combined with sorafenib. (**D**) Immunoblot analysis showing changes in SERCA and CHOP (ER stress markers) with nifedipine, KB-R7943, caloxin 2a1, and thapsigargin alone or combined with sorafenib. * *p* < 0.05 and ** *p* < 0.01 versus control. S, Sorafenib; N, Nifedipine; KB, KB-R7943; C, Caloxin2a1; T, Thapsigargin; (−), Sorafenib absence; (+), Sorafenib presence. Beta-actin served as a loading control.

**Figure 3 ijms-24-16413-f003:**
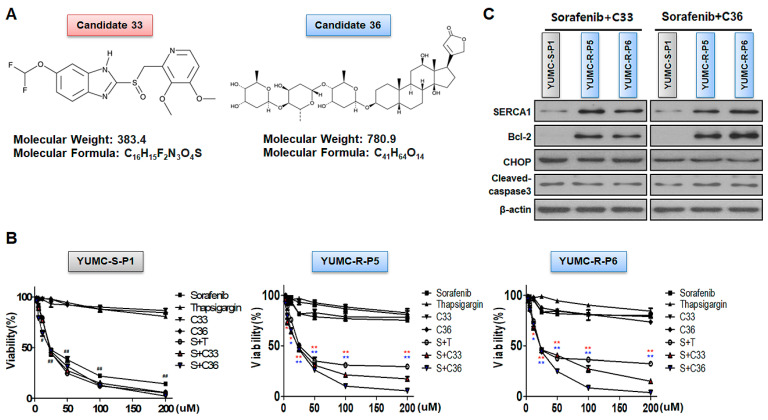
Discovery of novel SERCA inhibitors through in silico screening. (**A**) Chemical structures and information regarding novel SERCA inhibitors, candidates 33 and 36. (**B**) Cell viability between sorafenib-sensitive (left) and sorafenib-resistant (middle; YUMC-R-P5, right; -R-P6) PTC cell combinatorial strategy for novel SERCA inhibitors (candidates 33 and 36) and sorafenib. Points express the mean percentage of the values shown in the solvent-treated control. (**C**) Immunoblot analysis for combinational anti-cancer influence of sorafenib and novel SERCA inhibitors between sorafenib-sensitive and -resistant PTC cells. Beta-actin served as a loading control. All results were performed in triplicate or greater. The data represent the mean ± standard deviation. *^,#^ *p* < 0.05 and **^,##^ *p* < 0.01 versus control. *,** sorafenib + C33 versus control; *,** sorafenib + C36 versus control. S: Sorafenib; T, Thapsigargin; C33, Candidate 33; C36, Candidate 36. The red or blue asterisks were indicated effect of C33 or C36 compare than control respectively. The black asterisk (*) or sharp (^#^) indicated effect of sorafenib only treatment or combination with sorafenib and thapsigargin in YUMC-S-P1.

**Figure 4 ijms-24-16413-f004:**
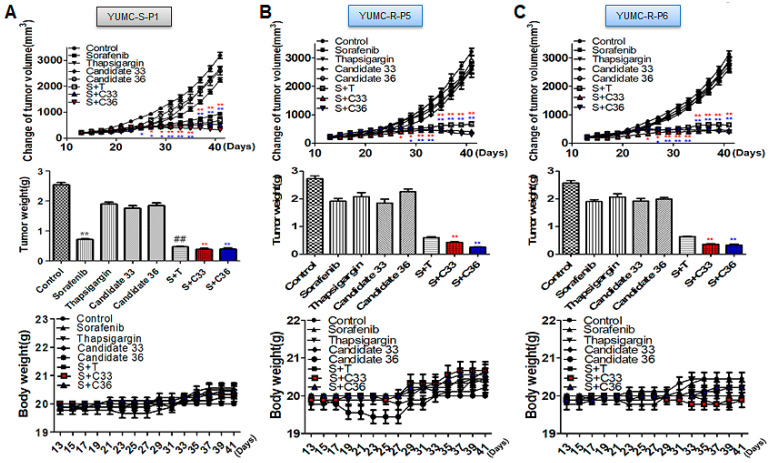
A combinatorial strategy for novel SERCA inhibitors and sorafenib significantly enhanced tumor shrinkage in a xenograft model with patient-derived sorafenib-resistant PTC cells. ((**A**–**C**), top), changes of tumor volume; ((**A**–**C**), middle) the resected tumor weight; ((**A**,**C**), bottom) changes in whole body weight (each group, *n* = 10). Tumor size was calculated in NOD/Shi-scid, IL-2Rγ KOJic (NOG) mice, and animals were treated with sorafenib combined with novel SERCA inhibitors (candidates 33 or 36) or with each agent alone. Data represent the mean ± standard error of the mean. * *p* < 0.05 and **^,##^ *p* < 0.01, compared with control. *,** sorafenib + C33 versus control; *,** sorafenib + C36 versus control. S: Sorafenib; T: Thapsigargin; C33: Candidate 33; C36: Candidate 36. The red or blue asterisks were indicated effect of C33 or C36 presence with sorafenib compare than control respectively. The black asterisk (*) or sharp (#) indicated effect of sorafenib only treatment or combination with sorafenib and thapsigargin in YUMC-S-P1.

**Figure 5 ijms-24-16413-f005:**
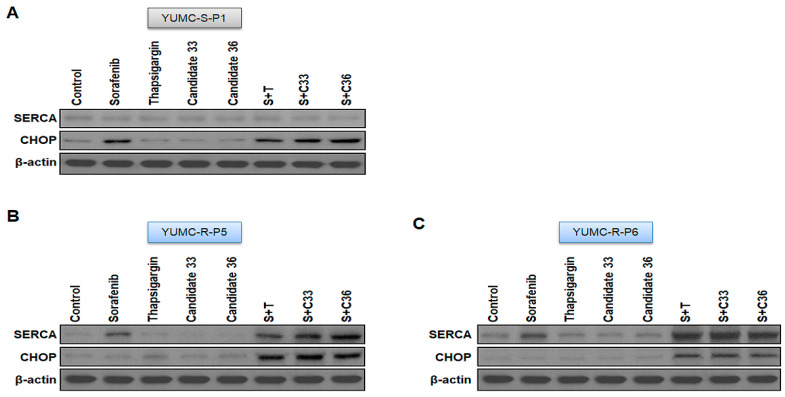
(**A**–**C**) Immunoblot analysis with SERCA and CHOP (endoplasmic reticulum stress marker) in a mouse xenograft model with sorafenib-sensitive (YUMC-S-P1) and -resistant (YUMC-R-P5 and -P6) PTC cells. Beta-actin served as a loading control.

**Table 1 ijms-24-16413-t001:** Properties and clinical features of patients. Patient-derived papillary thyroid cancer (PTC) cells were isolated from these patient specimens.

	YUMC-S-P1	YUMC-R-P5	YUMC-R-P6
Age at Diagnosis	53	52	57
Gender	Male	Female	Male
Primary Disease Site	Thyroid	Thyroid	Thyroid
Stage	T4aN1bM0	T4aN1bM1	T3N1bM1
Primary Pathology	Papillary thyroid cancer	Papillary thyroid cancer (Recurrence and metastasis after sorafenib treatment)	Papillary thyroid cancer (Recurrence and metastasis after sorafenib treatment)
Classification of specimen used for culture	Fresh tumor	Fresh tumor	Fresh tumor
Obtained from	Severance Hospital, Seoul, Republic of Korea	Severance Hospital, Seoul, Republic of Korea	Severance Hospital, Seoul, Republic of Korea

**Table 2 ijms-24-16413-t002:** IC_50_ values for the combinational value of SERCA inhibitors with sorafenib in sorafenib-sensitive and -resistant PTC cells. Each data point infers the mean of three particular MTT assays, carried out in triplicate. SEM, standard error of the mean; MTT, 3-(4,5-dimethylthiazol-2-yl)-2,5-diphenyltetrazolium bromide; IC_50_, half-maximal inhibitory concentration. S; Sorafenib, T; Thapsigargin, C33; Candidate 33, C36; Candidate 36.

Cell Line	Histopathology	Animal	Cell Proliferation IC_50_ (μM)			
			Sorafenib	S + T	S + C33	S + C36
YUMC-S-P1	Thyroid, Papillary	Human	22 (±0.2)	22 (±0.3)	22 (±0.1)	22 (±0.1)
YUMC-R-P5	Thyroid, Papillary	Human	–	22 (±0.1)	23 (±0.1)	23 (±0.3)
YUMC-R-P6	Thyroid, Papillary	Human	–	22 (±0.3)	23 (±0.2)	23 (±0.1)

## Data Availability

The data presented in this study are available on reasonable request from the corresponding author.

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
