# Peer review of "Discovery of New Anti-Cancer Agents against Patient-Derived Sorafenib-Resistant Papillary Thyroid Cancer"

_ijms, 2023, doi:10.3390/ijms242216413_

Round 1
Reviewer 1 Report
Comments and Suggestions for Authors
This is an interesting study describing some new anti-cancer agents for treating sorafenib-resistant papillary thyroid cancer. Although the manuscript needs extensive English editing, the study is well designed and has some interesting conclusions.
Major remark: extensive English revision. The manuscript requires extensive English corrections in order to be more comprehensible for the readers. There are too many grammar and typographical mistakes in all parts of the manuscript. Due to the mistakes, the manuscript is hard to follow in some parts.
Minor:
1. References num. 7, 8, 9, and 10 are not adequate. Please, revise
2. pg5: Instead of Fig2A, it should be Fig3A, and instead of Fig2B-C, it should be Fig3B-C.
3. pg10: The conclusion, “SERCA is a key regulator for calcium homeostasis under severe ER stress conditions via anti-cancer treatment,” is too strong and could not be concluded from your results. Please, revise
4. supplement - it is written: “Further protocol details are as described in our previous article [4].” But, the ref 4.(Yun, H.J.; Lim, J.H.; Kim, S.Y.; Kim, S.M.; Park, K.C. Discovery of pharmaceutical composition for prevention and treatment in patient-derived metastatic medullary thyroid carcinoma model. Biomedicines 2022, 10) includs also the line “Further protocol details are as described in our previous article [19].”
[19] Yun H.J., Kim M., Kim S.Y., Fang S., Kim Y., Chang H.S., Chang H.J., Park K.C. Effects of anti-cancer drug sensitivity-related genetic differences on therapeutic approaches in refractory papillary thyroid cancer. Int. J. Mol. Sci. 2022;23:699. doi: 10.3390/ijms23020699. [PMC free article] [PubMed] [CrossRef] [Google Scholar]
And, in the aforementioned article, it is written: Further details on the protocol can be found in our previous article [68].” and so long.
Please explain the protocol in this article without citing anything previous, as it seems that there is no beginning or ending.
Comments on the Quality of English LanguageThe manuscript requires extensive English editing in order to be more comprehensible for the readers. There are too many grammar and typographical mistakes in all parts of the manuscript. Due to the mistakes, the manuscript is hard to follow
Author Response
Response to Reviewer 1
This is an interesting study describing some new anti-cancer agents for treating sorafenib-resistant papillary thyroid cancer. Although the manuscript needs extensive English editing, the study is well designed and has some interesting conclusions.
> Reply: I don't know how to thank you enough for reviewing our manuscript. A point-by-point response to your expert comments was indicated below. Thank you again for your review. I hope you are always healthy and happy!!
Major remark: extensive English revision. The manuscript requires extensive English corrections in order to be more comprehensible for the readers. There are too many grammar and typographical mistakes in all parts of the manuscript. Due to the mistakes, the manuscript is hard to follow in some parts.
> Reply: Thank you for your comment. I follow your professional opinion, the entire draft has undergone thorough English revision.
Minor:
- References num. 7, 8, 9, and 10 are not adequate. Please, revise
> Reply: I have made corrections according to your expert opinion.
- pg5: Instead of Fig2A, it should be Fig3A, and instead of Fig2B-C, it should be Fig3B-C.
> Reply: I am sorry for the unnecessary confusion caused by my mistake. I have made corrections according to your expert opinion.
- pg10: The conclusion, “SERCA is a key regulator for calcium homeostasis under severe ER stress conditions via anti-cancer treatment,” is too strong and could not be concluded from your results. Please, revise
> Reply: Thank you for your comment. The sentence was deleted according to your professional opinion.
- supplement - it is written: “Further protocol details are as described in our previous article [4].” But, the ref 4.(Yun, H.J.; Lim, J.H.; Kim, S.Y.; Kim, S.M.; Park, K.C. Discovery of pharmaceutical composition for prevention and treatment in patient-derived metastatic medullary thyroid carcinoma model. Biomedicines 2022, 10) includs also the line “Further protocol details are as described in our previous article [19].” [19] Yun H.J., Kim M., Kim S.Y., Fang S., Kim Y., Chang H.S., Chang H.J., Park K.C. Effects of anti-cancer drug sensitivity-related genetic differences on therapeutic approaches in refractory papillary thyroid cancer. Int. J. Mol. Sci. 2022;23:699. doi: 10.3390/ijms23020699. [PMC free article] [PubMed] [CrossRef] [Google Scholar]
And, in the aforementioned article, it is written: Further details on the protocol can be found in our previous article [68].” and so long.Please explain the protocol in this article without citing anything previous, as it seems that there is no beginning or ending.
> Reply: The reference was modified in “Supplementary Table and Methods,” and method details were described in “Materials and Methods” instead of “Further details on the protocol can be found in our previous article” according to your expert opinion.
Reviewer 2 Report
Comments and Suggestions for Authors
The manuscript by Kim and collaborators focuses on novel inhibitor-based approach for treatment of sorafenib-resistant papillary thyroid cancer (PTC). The Authors propose the use of a SERCA-targeted inhibitor as an efficient therapeutic agent in a xenograft tumour model of patient-derived sorafenib-resistant PTC cells. The topic of the publication is interesting and should attract the attention of readers. Unfortunately, the manuscript needs significant editing for language and writing quality. Other suggested changes are outlined below.
[1] Introduction: It is worth mentioning that in 2022 WHO introduced a new classification of thyroid cancers neoplasms (e.g., Endocr Pathol. 2022 Mar;33(1):27-63). Additionally, I would recommend adding a reference to the article summarizing the current knowledge on inhibitor-based targeted therapies in thyroid cancer (Int J Mol Sci. 2021 Oct 31;22(21):11829).
[2] Figure 1A: Please use the current gene names provided in the GeneCards database (https://www.genecards.org/), for example, instead of MDR1, please use ABCB1. Similarly, MRP is a name of a group of at least several proteins, for example, MRP1, MRP2, MRP3, MRP6. Please specify which one you mean and use the correct name of the gene encoding it (e.g. ABCC1 for MRP1). Please make similar changes throughout the manuscript.
[3] Figures 1-5: It is not explained in the legend what the following abbreviations mean: S+N, S+KB, S+C, S+T, S+C33, S+C36, S+C24, S+C31.
[4] Please explain in section 2.3 why CHOP level was determined by Western blot technique (as shown e.g., in Fig. 2D). It is an ER stress marker, but this information only appears in the Figure 2 legend.
[5] Figure 2 legend: Please add the information in the legend to the Fig. 1 that beta-actin served as a loading control.
[6] Materials and Methods are briefly described in the main body of the manuscript, and in more detail in the Supplementary Materials. It seems to me that it is worthwhile to introduce such a description only in one place, that is, only in the main body of the manuscript.
[7] Please specify the suppliers and catalog number of the inhibitors used in the publication, i.e., nifedifine, KB R7943, caloxin2a1 and thapsigargin. It is not clear in what solvent each inhibitor was dissolved (DMSO?), if so at what concentration, and what the working concentration was used.
[8] In the Supplementary Materials (lines 63-64), the Authors state that isolated cancer cells were cultured in RPMI + 15% FBS without antibiotic/antimycotic agents. The lack of antibiotic/antimycotic surprises. Please comment.
[9] Figure 1A shows the bands for SERCA1, SERCA2, SERCA3 using the Western blot technique. Other Figures, e.g., 1D, 3D, 5, show bands signed generally SERCA (without a digit). Does this mean that these bands represent the expression of all these proteins? Please clarify.
[10] Materials and Methods lack information about the antibody that recognizes cleaved caspase 3. Please complete this.
[11] Supplementary Materials – Human PTC cell xenograft: Please state what the cells administered to the mice were suspended in. Explain “p.o.” abbreviation. Please write in what form sorafenib was administered (solid/liquid, if the latter - information on the diluent/volume per administration are needed).
[12] The descriptions of the X and Y axes in Figures 1 4 are not very clear.
Examples of typos:
[1] Page 2: “Noch […] signaling pathway” should be changed for “Notch pathway […] signaling pathway”. Similar change is needed on page 9.
[2] Figure 1 legend: “soorafenib-sensitive” should be changed for “sorafenib-sensitive”.
Comments on the Quality of English LanguageThe manuscript by Kim and collaborators focuses on novel inhibitor-based approach for treatment of sorafenib-resistant papillary thyroid cancer (PTC). The Authors propose the use of a SERCA-targeted inhibitor as an efficient therapeutic agent in a xenograft tumour model of patient-derived sorafenib-resistant PTC cells. The topic of the publication is interesting and should attract the attention of readers. Unfortunately, the manuscript needs significant editing for language and writing quality. Other suggested changes are outlined below.
[1] Introduction: It is worth mentioning that in 2022 WHO introduced a new classification of thyroid cancers neoplasms (e.g., Endocr Pathol. 2022 Mar;33(1):27-63). Additionally, I would recommend adding a reference to the article summarizing the current knowledge on inhibitor-based targeted therapies in thyroid cancer (Int J Mol Sci. 2021 Oct 31;22(21):11829).
[2] Figure 1A: Please use the current gene names provided in the GeneCards database (https://www.genecards.org/), for example, instead of MDR1, please use ABCB1. Similarly, MRP is a name of a group of at least several proteins, for example, MRP1, MRP2, MRP3, MRP6. Please specify which one you mean and use the correct name of the gene encoding it (e.g. ABCC1 for MRP1). Please make similar changes throughout the manuscript.
[3] Figures 1-5: It is not explained in the legend what the following abbreviations mean: S+N, S+KB, S+C, S+T, S+C33, S+C36, S+C24, S+C31.
[4] Please explain in section 2.3 why CHOP level was determined by Western blot technique (as shown e.g., in Fig. 2D). It is an ER stress marker, but this information only appears in the Figure 2 legend.
[5] Figure 2 legend: Please add the information in the legend to the Fig. 1 that beta-actin served as a loading control.
[6] Materials and Methods are briefly described in the main body of the manuscript, and in more detail in the Supplementary Materials. It seems to me that it is worthwhile to introduce such a description only in one place, that is, only in the main body of the manuscript.
[7] Please specify the suppliers and catalog number of the inhibitors used in the publication, i.e., nifedifine, KB R7943, caloxin2a1 and thapsigargin. It is not clear in what solvent each inhibitor was dissolved (DMSO?), if so at what concentration, and what the working concentration was used.
[8] In the Supplementary Materials (lines 63-64), the Authors state that isolated cancer cells were cultured in RPMI + 15% FBS without antibiotic/antimycotic agents. The lack of antibiotic/antimycotic surprises. Please comment.
[9] Figure 1A shows the bands for SERCA1, SERCA2, SERCA3 using the Western blot technique. Other Figures, e.g., 1D, 3D, 5, show bands signed generally SERCA (without a digit). Does this mean that these bands represent the expression of all these proteins? Please clarify.
[10] Materials and Methods lack information about the antibody that recognizes cleaved caspase 3. Please complete this.
[11] Supplementary Materials – Human PTC cell xenograft: Please state what the cells administered to the mice were suspended in. Explain “p.o.” abbreviation. Please write in what form sorafenib was administered (solid/liquid, if the latter - information on the diluent/volume per administration are needed).
[12] The descriptions of the X and Y axes in Figures 1 4 are not very clear.
Examples of typos:
[1] Page 2: “Noch […] signaling pathway” should be changed for “Notch pathway […] signaling pathway”. Similar change is needed on page 9.
[2] Figure 1 legend: “soorafenib-sensitive” should be changed for “sorafenib-sensitive”.
Author Response
Response to Reviewer 2
Comments and Suggestions for Authors
The manuscript by Kim and collaborators focuses on novel inhibitor-based approach for treatment of sorafenib-resistant papillary thyroid cancer (PTC). The Authors propose the use of a SERCA-targeted inhibitor as an efficient therapeutic agent in a xenograft tumour model of patient-derived sorafenib-resistant PTC cells. The topic of the publication is interesting and should attract the attention of readers. Unfortunately, the manuscript needs significant editing for language and writing quality. Other suggested changes are outlined below.
> Reply: I don't know how to thank you enough for reviewing our manuscript. A point-by-point response for your expert comments is indicated below. Thank you again for your review. I hope you are always healthy and happy!!
[1] Introduction: It is worth mentioning that in 2022 WHO introduced a new classification of thyroid cancers neoplasms (e.g., Endocr Pathol. 2022 Mar;33(1):27-63). Additionally, I would recommend adding a reference to the article summarizing the current knowledge on inhibitor-based targeted therapies in thyroid cancer (Int J Mol Sci. 2021 Oct 31;22(21):11829).
> Reply: Thank you for your comment. I have added a sentence in the “introduction” section and added the suggested references.
[2] Figure 1A: Please use the current gene names provided in the GeneCards database (https://www.genecards.org/), for example, instead of MDR1, please use ABCB1. Similarly, MRP is a name of a group of at least several proteins, for example, MRP1, MRP2, MRP3, MRP6. Please specify which one you mean and use the correct name of the gene encoding it (e.g. ABCC1 for MRP1). Please make similar changes throughout the manuscript.
> Reply: Thank you for your comment. I have made the suggested corrections to Figure 1A and the “Results” section.
[3] Figures 1-5: It is not explained in the legend what the following abbreviations mean: S+N, S+KB, S+C, S+T, S+C33, S+C36, S+C24, S+C31.
> Reply: I am sorry for the unnecessary confusion caused by my mistake. Abbreviations were added in the figure legends.
[4] Please explain in section 2.3 why CHOP level was determined by Western blot technique (as shown e.g., in Fig. 2D). It is an ER stress marker, but this information only appears in the Figure 2 legend.
> Reply: Thank you for your comment again. According to your experted opinion, I have added a sentence regarding the reason for determining “CHOP”.
[5] Figure 2 legend: Please add the information in the legend to the Fig. 1 that beta-actin served as a loading control.
> Reply: I have added the information explaining the loading control and normalization control in the legend of Fig 1, 2, 3, and 5.
[6] Materials and Methods are briefly described in the main body of the manuscript, and in more detail in the Supplementary Materials. It seems to me that it is worthwhile to introduce such a description only in one place, that is, only in the main body of the manuscript.
> Reply: Thank you for your comment. I have made corrections and added several sentences to the “Materials and Methods” section.
[7] Please specify the suppliers and catalog number of the inhibitors used in the publication, i.e., nifedifine, KB R7943, caloxin2a1 and thapsigargin. It is not clear in what solvent each inhibitor was dissolved (DMSO?), if so at what concentration, and what the working concentration was used.
> Reply: Thank you for your comment. I have made corrections and added several sentences to section 4.10 of “Materials and Methods.”
[8] In the Supplementary Materials (lines 63-64), the Authors state that isolated cancer cells were cultured in RPMI + 15% FBS without antibiotic/antimycotic agents. The lack of antibiotic/antimycotic surprises. Please comment.
> Reply: I am sorry for the unnecessary confusion caused by my mistake. I have made the corrections to section 4.4 of “Materials and Methods.”
[9] Figure 1A shows the bands for SERCA1, SERCA2, SERCA3 using the Western blot technique. Other Figures, e.g., 1D, 3D, 5, show bands signed generally SERCA (without a digit). Does this mean that these bands represent the expression of all these proteins? Please clarify.
> Reply: I am sorry for the unnecessary confusion caused by my mistake. I have made the recommended corrections to the figures.
[10] Materials and Methods lack information about the antibody that recognizes cleaved caspase 3. Please complete this.
> Reply: Thank you for your comment. I have added information regarding caspase 3 antibody in the “Materials and Methods” section.
[11] Supplementary Materials – Human PTC cell xenograft: Please state what the cells administered to the mice were suspended in. Explain “p.o.” abbreviation. Please write in what form sorafenib was administered (solid/liquid, if the latter - information on the diluent/volume per administration are needed).
> Reply: I have made the corrections to section 4.11 of “Materials and Methods” according to your expert opinion.
[12] The descriptions of the X and Y axes in Figures 1 4 are not very clear.
> Reply: Thank you for your comment. I have made corrections in Figure 1 and 4 according to your suggestions.
Examples of typos:
[1] Page 2: “Noch […] signaling pathway” should be changed for “Notch pathway […] signaling pathway”. Similar change is needed on page 9.
> Reply: Thank you for your comment. I have made corrections according to your suggestions.
[2] Figure 1 legend: “soorafenib-sensitive” should be changed for “sorafenib-sensitive”.
> Reply: Thank you for your comment. I have made corrections according to your suggestions.
Round 2
Reviewer 1 Report
Comments and Suggestions for Authors
Dear authors,
The revisions made to the manuscript have greatly enhanced its quality. Although there are still a few grammar mistakes in the revised version of the manuscript, the necessary changes have been implemented, resulting in a significantly improved version.
Comments on the Quality of English LanguageThere are still a few grammar mistakes and sentence syntax errors in the revised version of the manuscript, but the manuscript is sufficiently improved.
Author Response
The revisions made to the manuscript have greatly enhanced its quality. Although there are still a few grammar mistakes in the revised version of the manuscript, the necessary changes have been implemented, resulting in a significantly improved version.
> Reply: I have made corrections according to your expert opinion.
Reviewer 2 Report
Comments and Suggestions for Authors
The Authors addressed all my comments. I recommend the manuscript for publication.
Comments on the Quality of English LanguageThe Authors addressed all my comments. I recommend the work for publication. I would only recommend checking the article in terms of the quality of the English language used.
Author Response
The Authors addressed all my comments. I recommend the work for publication. I would only recommend checking the article in terms of the quality of the English language used.
> Reply: I don't know how to thank you enough for reviewing our manuscript.